# Morphological Characterization of Root System Architecture in Diverse Tomato Genotypes during Early Growth

**DOI:** 10.3390/ijms19123888

**Published:** 2018-12-05

**Authors:** Aurora Alaguero-Cordovilla, Francisco Javier Gran-Gómez, Sergio Tormos-Moltó, José Manuel Pérez-Pérez

**Affiliations:** 1Instituto de Bioingeniería, Universidad Miguel Hernández, 03202 Elche, Spain; aalaguero@umh.es (A.A.-C.); francisco.gran@goumh.umh.es (F.J.G.-G.); sertormol75@gmail.com (S.T.-M.); 2OQOTECH Process Validation System, 03801 Alcoy, Spain

**Keywords:** lateral root development, adventitious root development, plant phenomics, root growth analysis, root system architecture

## Abstract

Plant roots exploit morphological plasticity to adapt and respond to different soil environments. We characterized the root system architecture of nine wild tomato species and four cultivated tomato (*Solanum lycopersicum* L.) varieties during early growth in a controlled environment. Additionally, the root system architecture of six near-isogenic lines from the tomato ‘Micro-Tom’ mutant collection was also studied. These lines were affected in key genes of ethylene, abscisic acid, and anthocyanin pathways. We found extensive differences between the studied lines for a number of meaningful morphological traits, such as lateral root distribution, lateral root length or adventitious root development, which might represent adaptations to local soil conditions during speciation and subsequent domestication. Taken together, our results provide a general quantitative framework for comparing root system architecture in tomato seedlings and other related species.

## 1. Introduction

The root system is essential for plant growth because of its basic functions in the selective absorption of water and nutrients, as a mechanical support and storage organ, as a selective barrier against pathogens, and in the modulation of some stress responses [1,2]. However, our knowledge about the genetic mechanisms that modulate root system architecture (RSA) in species of agronomic interest is, with some exceptions, very limited [3,4,5]. Cultivated tomato (*Solanum lycopersicum* L.) is an important vegetable grown worldwide [6]. Tomato crops are particularly sensitive to drought, have poor nitrogen and phosphorus use efficiency and consequently require intensive irrigation and fertilization to maintain high yield and fruit quality [7,8]. Manipulation of RSA traits may improve water and nutrient capture under normal and extreme climate conditions [9,10]. The cultivated tomato is phylogenetically related to another 13 species of wild tomatoes, all of which are native to South America and show considerable morphological and ecological diversity [11]. Compared with the large genetic variability found in wild tomato species, the genetic diversity of the thousands of cultivated tomato varieties is very limited due to their recent domestication from a small number of individuals [12,13].

A detailed characterization of root development during early growth in two related tomato species, *S. pennellii* and *S. lycopersicum* ‘M82’, has been performed previously, which provided significant differences in a large range of root traits with developmental significance [14]. Additionally, a quantitative analysis of cellular and morphological root phenotypes in a population of 76 homozygous introgression lines between these two species featured numerous quantitative trait loci that influence a diversity of root traits [14]. Further analysis of this population can facilitate the eventual identification of genes that regulate some key RSA attributes, such as root length. Conversely, few studies have employed a genetic approach to examine the role of specific factors on tomato RSA. Previous studies identified an essential role for the *DIAGEOTROPICA* (*DGT*) gene in the development of lateral roots (LRs) in tomato [15]. *DGT* encodes a cyclophilin protein that negatively regulates PIN-FORMED auxin efflux transporters by affecting their plasma membrane localization; hence, the *dgt* mutant lacked the auxin maxima relevant to the priming and specification of LR founder cells and was consequently impaired in LR organogenesis [16]. In another study, the regulation of tomato RSA by flavonols was revealed by the analysis of the *anthocyanin reduced* (*are*) mutant, which has been suggested to have a defect in the gene encoding the enzyme FLAVONOID 3-HYDROXYLASE [17]. Auxin transport enhancement, alterations in auxin-induced gene expression and reduced LR initiation in these mutants are consistent with flavonols, reducing auxin transport through the wild-type roots and driving the accumulation of auxin at sites of LR primordia formation [17].

To characterize the phenotypic space of RSA in tomato, we studied several morphological traits during early growth in 19 tomato genotypes selected from a representative sample of wild tomato species, commercial cultivars and monogenic mutants. One the one hand, differential RSA traits among commercial tomato cultivars and related wild tomato species might represent adaptations to local soil conditions that could have been positively selected during domestication or, alternatively, that these traits were genetically linked to the yield-associated traits selected during domestication. On the other hand, the characterization of early RSA traits in a number of developmental mutants of the same genetic background will allow us to understand the hormonal crosstalk contributing to the local activation of growth in postembryonic root meristems. Our results will provide a theoretical framework to initiate the genetic characterization of RSA in tomato seedlings during early growth.

## 2. Results and Discussion

We established a precise in vitro experimental setup (Appendix A) to explore the RSA of different tomato genotypes (Table 1) during their early growth. Based on the recent phylogeny of wild tomato species (section *Lycopersicon*) [11,12,13], we selected nine tomato relatives and four reference commercial cultivars for our studies. To search for novel RSA regulators, we also investigated some of the developmental mutants that were introgressed previously by other authors into a unique background, the ‘Micro-Tom’ cultivar [18], that facilitates comparative studies and double mutant analysis.

### 2.1. Germination and Early Root Growth

The studied genotypes showed a quick germination as most seeds germinated between 24 and 72 h on wet chamber incubation. *S. chmielewskii*, *S. arcanum* and *S. cheesmaniae* displayed a slight delay in germination, which otherwise did not result in a reduced germination percentage at sowing time (Appendix A). The primary roots (PRs) of ‘Moneymaker’ and ‘Ailsa Craig’ grew at higher rates than those of ‘Craigella’ and ‘Micro-Tom’ during the first three days of growth on plates (Figure 1a,b). Regarding the wild tomato species analyzed, *S. chmielewskii* displayed the lowest PR growth rate (Figure 1a,b). Some of the studied mutants, such as *anthocyanin absent* (*aa*), also showed a delay in germination compared to their counterparts in the ‘Micro-Tom’ background (Appendix A), indicating a putative role of the *AA* gene in root emergence. Additionally, the *bushy* mutants displayed shorter PR lengths (5.53 ± 2.61 mm; *n* = 18) compared to their counterparts in the ‘Micro-Tom’ background (11.93 ± 3.51 mm; *n* = 20; Figure 1a and Appendix A), which is already affected by cell expansion due to recessive mutations leading to brassinosteroid deficiency [19].

Our studies on different tomato genotypes and wild relatives indicated that observed differences during early growth were caused by a combination of (i) differences in their germination time caused by delays in PR protrusion and (ii) growth rate differences of the emerged PRs. In most wild tomato relatives, both PR emergence and PR growth rates were reduced compared to commercial tomato cultivars and their ancestor *S. pimpinellifolium*. Despite the close phylogenetic relationship between *S. chmielewskii* and *S. arcanum* [11,12], both species differed significantly in PR growth rates (Figure 1b), which may indicate adaptation to local soil conditions. The uneven germination of wild tomato species provides an adaptive advantage to rapidly changing environmental conditions (i.e., soil moisture), allowing higher seedling survival; however, uniform germination and rapid seedling growth are prerequisites for crop species [24] that might have been positively selected during tomato domestication [25]. We also found striking differences in early growth rates between commercial cultivars ‘Moneymaker’ and ‘Craigella’ that deserve further investigation.

### 2.2. Lateral Root (LR) Capacity Assay

Previous studies in maize reported significant variations in key RSA parameters, such as LR density, LR length and LR growth angle, between different genotypes [26,27]. Additionally, extensive variations in these parameters have been observed across environments [28,29,30,31]. Hence, the dynamic modulation of RSA through time by defined genotype × environment interactions determines root plasticity responses and allows plants to efficiently adapt to environmental constraints [10,30,32].

To characterize RSA during early growth in tomato (see Materials and Methods), we measured several traits (Appendix A) in the newly emerged LRs three days after root tip excision (dae), which is known to promote rapid growth of already-specified and dormant LR primordia in Arabidopsis [33]. The number of newly emerged LRs was positively and significantly correlated with PR length at root tip excision in wild tomato relatives and commercial cultivars (Appendix A), which was consistent with the results found in other species [34,35,36]. *S. pimpinellifolium* showed the highest number of LRs, *S. peruvianum* displayed non-significant differences in the number of LRs compared to commercial cultivars, and the other wild tomato relatives showed lower LR numbers (Figure 2a,b and Appendix A). LRs were not evenly distributed along the PR length, with the lowest frequencies found in the distal end of the PR in most genotypes (Appendix A); in *S. arcanum* and *S. galapagense,* however, LRs were found with lower frequencies in the proximal region of the PR close to the hypocotyl base (Figure 2c). The average distances between LRs were significantly different among the studied genotypes (Figure 2d). *S. peruvianum*, *S. chmielewskii* and ‘Craigella’ displayed the smallest average distances between adjacent LRs; in contrast, the LR average distances in *S. chilense* and *S. arcanum* were almost doubled (Figure 2d).

LRs display growth tropisms in response to gravity, light, touch and moisture gradients that contribute to enhancing plant growth by increasing nutrient capture from the soil [37]. Although LR distribution in Arabidopsis has been linked to the waving growth pattern of the PR through asymmetric auxin accumulation [38], the LRs in tomato did not show a preference to growing on the outside edge of the PR (Appendix A). For most genotypes, ~60% of the LRs emerged in alternating directions along the PR length, and approximately 20–30% of the LRs emerged from the same edge of the PR in clusters of two to three LRs (Appendix A). In *S. peruvianum*, *S. huaylasense* and *S. pimpinellifolium,* there was a substantial proportion of seedlings (~50%) with clusters of four or more LRs that emerged on the same side of the PR. In monocot plants, such as rice and wheat, a strong correlation between the root growth angle and drought tolerance has been observed where steeper growth angles increased water capture [39,40]. In our tomato population, the average LR growth angle was 113.9 ± 17.5° (*n* = 1429), with extreme values found for *S. arcanum* and ‘Moneymaker’ with 106.4 ± 10.3° (*n* = 69) and 123.5 ± 20.5° (*n* = 102), respectively (Appendix A). Drought tolerant rice cultivars bearing functional alleles of *DEEPER ROOTING 1* (*DRO1*) develop LRs with steeper growth angles, reaching deeper into the soil, which maintained high yield performance under water deficit regimes [41]. As *DRO1*-related genes also influence RSA in dicot plants, such as Arabidopsis and plum [42], it is plausible that the observed differences in LR growth angle between tomato genotypes might be explained by natural variation in the DRO1 pathway as well.

In dicot plants, such as Arabidopsis and tomato, the formation of LRs occurs from a subset of pericycle cells that are periodically primed at the so-called oscillation zone of the PR [38,43]. In these species, LRs emerge on a basipetal pattern: LR outgrowth is initiated in a more proximal region of the PR, and hence, the older LRs are longer than the newly emerged LRs [44,45]. LR lengths in our experimental dataset approach a gamma distribution (Appendix A), which might reflect the temporal delay between the early steps of LR initiation and subsequent LR outgrowth after meristem activation. We found significant differences in the distribution of LR length between wild tomato species and commercial cultivars (Appendix A). Most commercial cultivars and *S. peruvianum* displayed longer LRs (Figure 2e), while the LRs in *S. corneliomulleri*, *S. chmielewskii*, *S. arcanum* and *S. galapagense* were much shorter (Figure 2e). The RSA, which is defined by the length of the PR and the distribution, density, length and growth angle of the LRs, determines the soil volume that is explored by a single plant. Additionally, the high degree of plasticity of the root systems allows postembryonic alterations to occur in response to local environmental cues, such as nutrient deficiencies in the soil [30,31,46]. Indeed, a strong shift from PR growth to LR growth is observed in response to phosphate deficiency, which leads to a shorter PR with a high number of longer LRs [30,47,48]. We found substantial variation of early RSA parameters in a small selection of wild tomato species and commercial cultivars, which might represent local adaptation to soil conditions during speciation and that could have been selected during tomato domestication.

Most of the developmental mutants studied in this work displayed altered RSA after root tip excision compared with their counterparts in the ‘Micro-Tom’ background (Figure 3 and Appendix A). In contrast to ‘Micro-Tom’, we found a non-significant correlation between LR number and PR length in *lutescent* and *sitiens* (*sit*) mutants (Figure 3b), which also displayed a reduced number of LRs after root tip excision (Figure 3c), indicating that these mutants were affected in LR specification (i.e., prepatterning) and/or LR initiation. Other mutants with significantly lower numbers of LRs than ‘Micro-Tom’ were *aa*, *bushy* and *Never ripe* (*Nr*) (Figure 3c). By comparing the spatial distribution of LRs along the main root in the studied mutants (Appendix A), we found that most LRs in *aa* and *sit* mutants emerged in the PR region nearest the hypocotyl (Figure 3d), which indicated a substantial delay in LR initiation from the distal region of the PR or a failure to initiate new LRs after root tip excision in these mutants. Interestingly, LR density, estimated as the average distance between two consecutive LRs, was similar in all the studied mutants (Figure 3e). Accumulating evidence suggests that abscisic acid (ABA) plays an important role in stress-regulated root growth suppression [49]. Additionally, studies with ABA-deficient mutants indicate that ABA promotes stem cell maintenance in the root meristem [50]. Consistent with the latter, the ABA-deficient *sit* mutant, which is blocked in the conversion of ABA-aldehyde to ABA [23], displayed a reduced root system during early growth, indicating a requirement of ABA in LR initiation.

The LR lengths were significantly shorter in *Nr* mutants (2.59 ± 1.59 mm; *n* = 107) compared to ‘Micro-Tom’ (3.62 ± 2.39 mm; *n* = 306), while the anthocyanin-defective mutants studied, *aa* and *anthocyaninless* (*a*), displayed significantly longer LRs (Figure 3f). Among the studied mutants, *Nr* displayed more LR emerging on the same side of the PR than ‘Micro-Tom’, as well as steeper LR growth angles (Appendix A).

We studied RSA during the early growth of two mutants with reduced levels of anthocyanins. A previous study identified positive roles for anthocyanins in LR formation in tomato, likely through their direct regulation of polar auxin transport and the levels of reactive oxygen species [17]. The *a* mutant bears a frameshift mutation in a gene encoding the flavonoid 3′,5′-hydroxylase involved in the conversion of dihydrokaempferol to dihydromyricetin [21]. Recently, the deletion of a gene encoding a putative glutathione S-transferase (*SlGSTAA*) has been proposed as the causal mutation in the *aa* allele [20]. Interestingly, the strongest phenotype observed for RSA in anthocyanin-deficient mutants during early growth corresponded to the *aa* mutant. SlGSTAA is homologous to Arabidopsis TRANSPARENT TESTA19, which functions as a carrier to transport and sequester anthocyanins into the vacuole [51]. Another mutant with reduced anthocyanin levels also displayed altered RSA [17] similar to that found in the *aa* mutants, and the mild phenotype shown in RSA during early growth for the *a* mutant might be due to the different steps where the anthocyanin function is affected in each of these mutants.

### 2.3. Adventitious Root (AR) Formation in the Hypocotyl after Wounding

ARs are postembryonic roots that are formed from non-root tissues, such as leaves and stems, naturally or in response to altered environments [52,53]; these structures may also be induced by mechanical damage or during vegetative propagation of stem cuttings [54,55]. To test the ability of the different tomato genotypes to produce ARs after wounding, we excised the whole root system 6 days after sowing (das; Appendix A), which induced AR formation at the hypocotyl base above the wounding site shortly afterwards (Figure 4a and Appendix A). We followed AR initiation by scoring the presence of newly emerged AR primordia at the hypocotyl (see Materials and Methods). We found a significant delay in AR initiation in *S. cheesmaniae* and *S. chilense* compared to that in other studied wild tomato species and commercial cultivars; the emergence rate of consecutive ARs also varied, although not significantly, among most of the studied genotypes (Appendix A). We found striking differences in the number of ARs produced by the studied genotypes at 6 and 10 days after induction (dai), irrespective of their hypocotyl length (Figure 4b). Interestingly, the four studied commercial cultivars produced more ARs (7.0 ± 1.6 ARs at 10 dai; *n* = 102) than the other wild tomato species (Figure 4c), including the direct ancestor of cultivated tomato, *S. pimpinellifolium* (3.3 ± 1.2 ARs at 10 dai; *n* = 21), suggesting that enhanced AR formation could have been positively selected during domestication or, alternatively, that this trait was genetically linked to the other yield-associated traits selected during *S. lycopersicum* domestication [56]. AR growth rates were significantly higher in commercial cultivars, *S. chmielewskii*, *S. galapagense* and *S. pimpinellifolium*, than in the other studied genotypes (Figure 4d). In contrast, we found no differences in the growth rates of the first, second and third ARs during the first 12 h for the studied genotypes (Figure 4d), suggesting that ARs grow autonomously from the hypocotyl after emergence or that the hypocotyl-derived signal fueling AR growth was not limiting. Differences in AR length were observed between the wild tomato species and commercial cultivars studied at 6 dai (Figure 4e), but these differences were normalized at later stages (Appendix A), indicating a genotype-dependent dynamic regulation of the AR growth rate. Concerning the growth angle of ARs, *S. arcanum* and ‘Moneymaker’ also presented extreme values, with 110.8 ± 8.2° (*n* = 29) and 131.4 ± 24.1° (*n* = 33), respectively (Appendix A), which were very similar to the growth angles found for their LRs (see above).

When compared to other commercial cultivars, some traits of the AR system in ‘Micro-Tom’, such as AR initiation, AR growth rate, AR growth angle and total AR length, were not significantly different among the studied genotypes (Figure 4 and Appendix A). The number of ARs, however, was significantly reduced in the ‘Micro-Tom’ cultivar (3.2 ± 0.8 ARs at 10 dai; *n* = 25), which was correlated with its smaller hypocotyl length (Figure 4b–c). Regarding the studied mutants in the ‘Micro-Tom’ background, the *Nr* mutants showed some delay in AR initiation and in the emergence rate of consecutive ARs, while the *aa* and *sit* mutants displayed a significant reduction in AR initiation (Appendix A). Consistent with these results, ARs were longer in *aa* and *sit* mutants than in *Nr* mutants (Figure 4E and Appendix A). Ethylene and ABA have a complicated interaction in many developmental processes [57]. In Arabidopsis and tomato, ethylene caused a reduction in PR elongation and inhibited the initiation and elongation of LRs [58,59]. Interestingly, the results found in tomato suggested a positive role for ethylene in the regulation of ARs through the modulation of auxin transport [60,61]. In addition, submerged tomato roots triggered ethylene synthesis, which is required for flooding-induced auxin accumulation in the hypocotyl and thus AR formation [61]. Our results are consistent with *Nr* being affected in both the local activation of LR growth after root tip excision and AR initiation in the hypocotyl after whole root excision, which directly indicates a role for ethylene in wound-induced postembryonic root formation. In a recent study, transcriptome analysis during flood-induced AR formation in *Solanum dulcamara* uncovered a tissue-specific crosstalk between ethylene and ABA levels [62]. According to this model, the flooding-dependent ethylene response pathway that activates AR formation controls two downstream processes: the suppression of ABA signaling and the enhancement of auxin signaling [62]. The observed AR phenotypes in *Nr* and *sit* mutants suggest the function of a similar pathway during wound-induced AR formation, which will require further investigation.

### 2.4. Capturing Quantitative Variation of Early RSA in Tomato

To generate a parametrized space that captured variation in early RSA for the studied genotypes, we performed a heatmap representation with some of the parameters measured (Figure 5a). The studied commercial cultivars shared similar RSA traits and were clustered together and with its direct ancestor, *S. pimpinellifolium*. Although *S. chmielewskii*, *S. galapagense* and *S. cheesmaniae* clustered together, these plants displayed contrasting differences in some of the studied traits, such as PR growth rate and AR growth rate, indicating that these two traits might be independently controlled. Interestingly, some of the mutants in the ‘Micro-Tom’ background displayed specific alterations in some of the studied traits, such as LR and AR length (*aa*), which are likely controlled by the same genetic pathway. To further define the morphological space for early RSA in tomato, we chose six relevant RSA traits, PR growth, LR number, LR distance, LR distribution, LR length and LR growth angle, as well as four additional traits associated with the wound-induced AR system, number, length, growth angle and growth rate. For each of these traits, we defined three distinct states, named A, B and C, representing the first quartile, the second plus the third quartile and the fourth quartile of measured data values, respectively. The studied genotypes were grouped according to these traits and states (Appendix A), and representative diagrams of the early RSA ideotypes found in our study were drawn (Figure 5b).

Plant roots exploit morphological plasticity to adapt and respond to different soil environments and therefore help to improve resource use efficiency and to maintain productivity during limited nutrient and water availability [9,63]. Genotypic variation for RSA traits has been reported for many crops [64,65,66], and thus represents a suitable toolbox for targeted breeding. In our study, we identified a number of meaningful traits in tomato RSA (such as LR distribution in the soil depth, LR length or AR number) that displayed significant variation in a reduced number of commercial cultivars and related wild species. We plan to extend our root phenotyping studies to additional wild tomato accessions and commercial cultivars and to test their performance in response to some nutrient (phosphate and nitrate) deficiencies as well as simulated drought. Recently, a number of noninvasive methods for imaging plant roots on natural substrates have been developed [67,68], but their use is restricted due to high infrastructure cost or the availability of specific expertise. A low-cost high-throughput system for tomato root phenotyping has been developed in our lab, allowing automated image capture and implementation of software tools for root tracing and data analysis. To identify the genetic determinants of some of the differences found in RSA in tomato, we are studying a recombinant inbred line population derived from the cross *Solanum lycopersicum* ‘Moneymaker’ × *Solanum pimpinellifolium* [69]. The characterization of early RSA traits in the developmental mutants studied here indicated crosstalk between ethylene and abscisic acid signals for the local activation of growth in postembryonic root meristems. Additionally, several mutant collections on the ‘Micro-Tom’ background are available [70,71,72,73], which will allow screening for novel regulators of RSA in tomato and the rapid identification of the causal mutations by using a mapping-by-sequencing approach [74]. Our results provide a theoretical framework to initiate the genetic characterization of early RSA in tomato.

## 3. Materials and Methods

### 3.1. Plant Material and Growth Conditions

Seeds of wild tomato species and cultivated tomato (*Solanum lycopersicum* L.) varieties (Table 1) were obtained from the C.M. Rick Tomato Genetics Resource Center (TGRC; http://tgrc.ucdavis.edu/). The tomato cultivar ‘Micro-Tom’ and the near-isogenic lines (NILs) used in this study (Table 1) were obtained from the tomato mutant collection [18] maintained at the Escola Superior de Agricultura “Luiz de Queiroz”, Universidade de São Paulo, Brazil (http://www.esalq.usp.br/tomato/). Seeds of cultivated tomato cultivars and ‘Micro-Tom’ mutants were harvested between March and July 2017 from healthy plants grown in a gothic arch greenhouse at 38°16′43″ N, 0°41′15″ W, and 96 m altitude (Elche, Spain).

The seeds were surface-sterilized in 2% weight/volume sodium hypochlorite for 10 min, rinsed thoroughly with sterile distilled water (4 times) and cold-stored for 2 days. The sterilized seeds were transferred to wet chambers at 28 °C in a dark growth cabinet for 96 h. Germination was monitored daily on a sample of approximately 50 seeds per genotype. After 72 h on the wet chambers, germinated seedlings with a radicle >2 mm were transferred to 120 mm-square Petri dishes (0 days after sowing; das) containing 75 mL of sterile half-Murashige and Skoog basal salt medium (Duchefa, The Netherlands), 20 g L^−1^ sucrose (Duchefa), 2.5 g L^−1^ Gelrite (Duchefa), 0.5 g L^−1^ 2-(N-morpholino) ethanesulfonic acid (MES; Duchefa) and 2 mL L^−1^ Gamborg B5 vitamin solution (Duchefa), pH 5.8. Seven germinated seedlings were placed on each Petri dish, and three to five dishes per genotype were maintained in nearly vertical positions in a growth cabinet during 16 h light (average photosynthetic photon flux density of 50 µmol m^−2^ s^−1^) at 26 ± 1 °C and 8 h darkness at 23 ± 1 °C. In the lateral root (LR) capacity assay [33], 3–4 mm of the root tip was excised at 3 das, and the seedlings were grown for another 3 days (Appendix A); newly emerged LRs were then counted under a dissecting microscope. The formation of hypocotyl-derived adventitious roots (ARs) was induced at 6 das by removing the entire root system of each plant 1-2 mm above the hypocotyl-root junction with a sharp scalpel; ARs were periodically counted between 12 and 18 das (6 and 12 days after AR induction [dai], respectively). Primary root (PR), LR and AR pictures were taken using a Sony Cyber-shot DSC-H3 camera (Sony Corporation, Tokyo, Japan) at a resolution of 3264×2448 pixels and saved as an RGB color image in the jpeg format.

To measure AR initiation time, AR growth rate and AR growth angle, 4 dai (10 das) seedlings were transferred to new nearly vertically oriented plates, and serial images were taken every 3 h using a Canon EOS1100D camera (Canon Inc., Tokyo, Japan) with a Canon EF-S 17–55 mm f/3.5–5.6 lens at a resolution of 4272 × 2848 pixels and saved as an RGB color image in the jpeg format.

### 3.2. Image Analysis

PR length was estimated by the “Measure” tool after drawing a segmented line along the main root using Fiji [75]. For the measurement of most RSA traits (Appendix A), we used the EZ-Rhizo software as described elsewhere [76]. Briefly, all roots from a single image were semi-automated processed and skeletonized and then manually corrected for reconnecting any discontinuities in the root path and/or for separating overlapping roots. All measured RSA parameters in a text file were exported to an Excel datasheet (Appendix A). AR initiation and AR response were estimated by visually screening time-series images for the emergence of AR at the hypocotyl base in each seedling. The maximum AR length was measured in the AR system of seedlings 10 dai (18 das) by using a ruler.

### 3.3. Statistical Analyses

Descriptive statistics (average, standard deviation [SD], median, maximum and minimum) were calculated by using the StatGraphics Centurion XV software (StatPoint Technologies, Inc. Warrenton, VA, USA) and SPSS 21.0.0 (SPSS Inc., Chicago, IL, USA) programs. Data outliers were identified based on aberrant SD values and excluded for posterior analyses as described elsewhere [77]. One-sample Kolmogorov–Smirnov tests were performed to analyze the goodness-of-fit between the distribution of the data and a theoretical distribution (normal, gamma, log-logistic, or Weibull). Correlations were studied using Pearson product-moment correlation coefficient (Pearson’s r). Non-normal data values were transformed before the ANOVA by using $\sqrt{x}$ (length, distance), $\log_{2}x$ (growth angle) or $\frac{1}{x}$ (initiation time). Average ± SD values were represented, except in cases that did not exhibit a normal distribution and for which the median was used instead. We performed multiple testing analyses using the ANOVA F-test or Fisher’s least significant difference (LSD) methods (*p*-value < 0.01). Nonparametric tests were used when necessary.

### 3.4. Heat Map Representation

Standardized datasets obtained from the analysis of early RSA were processed using the pheatmap package of R version 3.3.2 (http://www.r-project.org/). Euclidean distance matrixes between morphological parameters (rows) and genotypes (columns) were calculated to build the dendrograms.

## Figures and Tables

**Figure 1 ijms-19-03888-f001:**
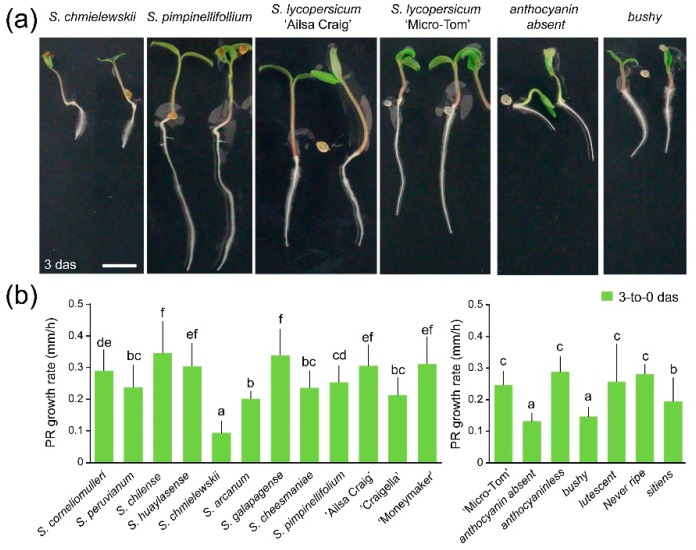
Early growth variation of selected tomato genotypes. (**a**) Seedlings of tomato genotypes differing in early root growth. Scale bar: 10 mm. (**b**) Growth rate (mm/h) of PRs in wild tomato species and commercial tomato cultivars (left), as well as in developmental mutants in the ‘Micro-Tom’ genetic background (right). Average ± SD values are shown. Different letters indicate significant differences (LSD; *p*-value < 0.01) over genotypes; das: days after sowing.

**Figure 2 ijms-19-03888-f002:**
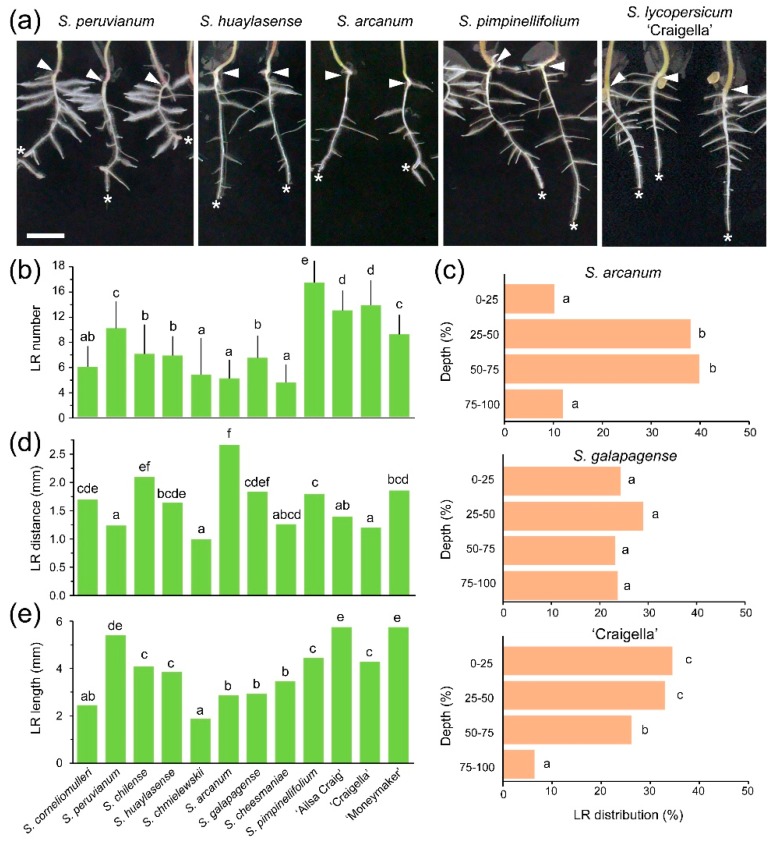
LR development in wild tomato species and commercial tomato cultivars after root tip excision. (**a**) Representative images of the entire root system of selected genotypes at 3 days after root tip excision. The arrowhead points to the root-hypocotyl junction, and the asterisk indicates the tip of the PR. Scale bar: 10 mm. (**b**) Number of LRs. Average ± SD values are shown. (**c**) Average percentages of LR distribution along the length (i.e., depth) of the PR. (**d**) Distance between consecutive LRs and (**e**) LR length; median values are shown. Different letters indicate significant differences (LSD; *p*-value < 0.01) over genotypes (b,d,e) or regarding PR depth domains (c).

**Figure 3 ijms-19-03888-f003:**
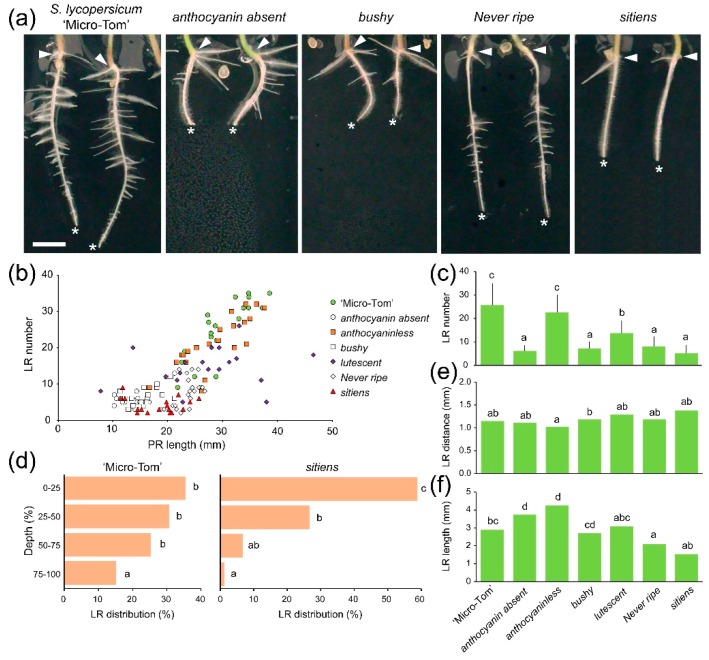
LR development in developmental mutants after root tip excision. (**a**) Representative images of the entire root system of selected genotypes 3 dae. The arrowhead points to the root-hypocotyl junction, and the asterisk indicates the distal tip of the PR. Scale bar: 10 mm. (**b**) Scatter plot of the LR number according to PR length. (**c**) Number of LRs. Average ± SD values are shown. (**d**) Average percentages of LR distribution along the length (i.e., depth) of the PR. (**e**) Distance between consecutive LRs and (**f**) LR length; median values are shown. Different letters indicate significant differences (LSD; *p*-value < 0.01) over genotypes (c,e,f) or regarding PR depth (d).

**Figure 4 ijms-19-03888-f004:**
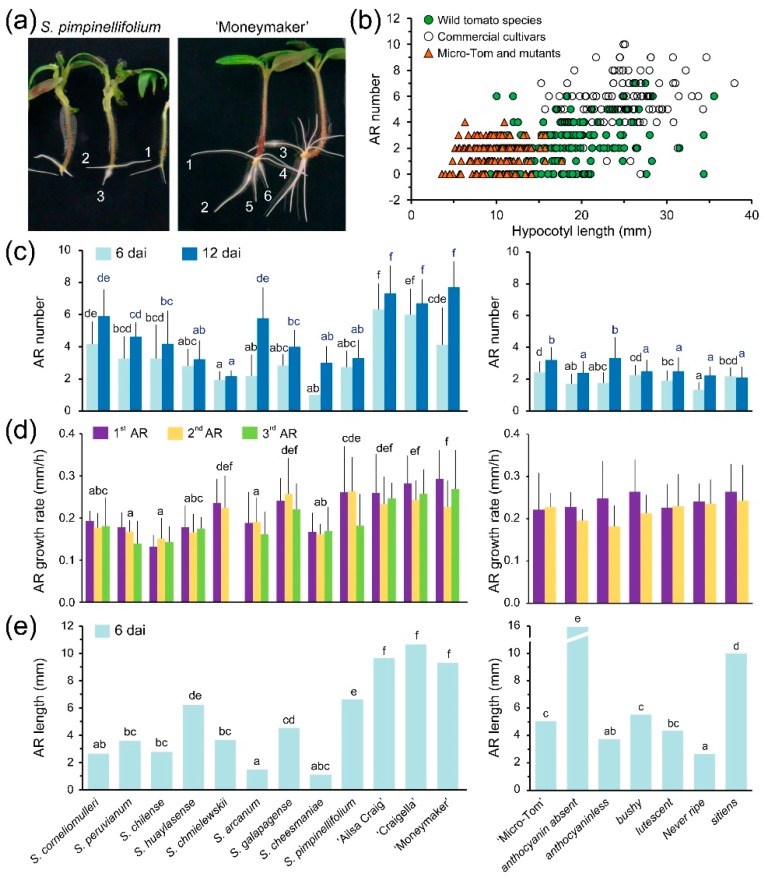
Variation of some AR traits in the studied genotypes after whole root excision. (**a**) Representative images of the AR system of selected genotypes at 10 dai. The numbers indicate the order of emergence of consecutive ARs. (**b**) Scatter plot of AR number according to hypocotyl length at 6 dai. (**c**) Number of Ars at 6 and 10 dai. Average ± SD values are shown. (**d**) AR growth rate of the first, second and third ARs during the first 12 h post-emergence. (**e**) Median values of AR length. Different letters indicate significant differences (LSD; *p*-value < 0.01) over genotypes (c–e).

**Figure 5 ijms-19-03888-f005:**
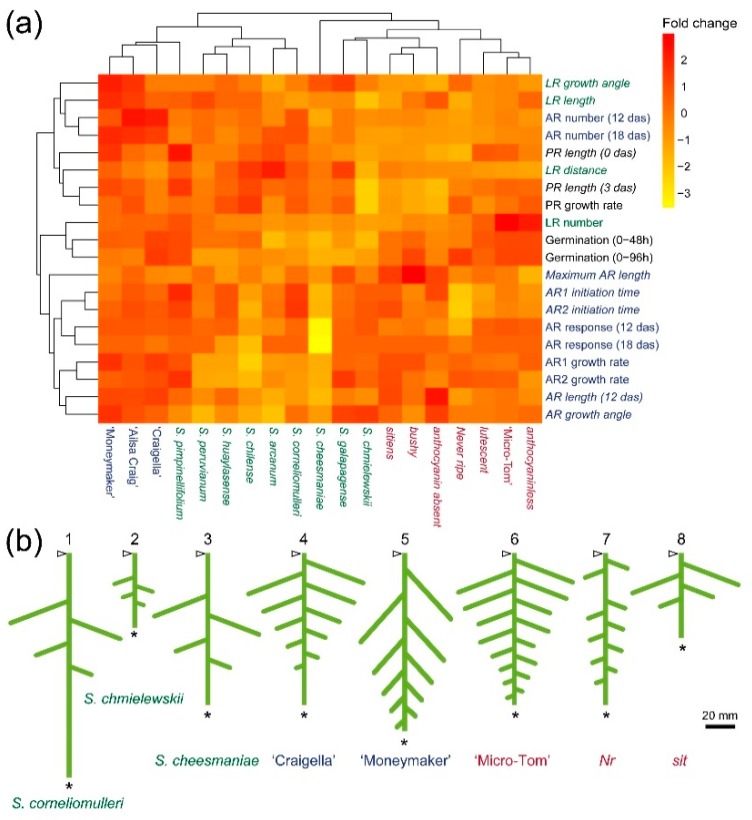
Morphological characteristics of early RSA in tomato. (**a**) Heat map representation of wild tomato species (green), commercial tomato cultivars (blue) and ‘Micro-Tom’ developmental mutants (red). Some of the morphological parameters analyzed (black, germination and early root growth; green, LR capacity assay; blue, wound-induced AR formation) are shown in the right column. The data from the parameters indicated in italics were transformed before the analysis (see Materials and Methods). The color code in the histogram ranges from yellow (lowest values) to red (highest values). (**b**) Early RSA ideotypes found in the studied tomato genotypes. The arrowhead points to the root-hypocotyl junction, and the asterisk indicates the tip of the PR. Representative genotypes for each ideotype are indicated.

**Table 1 ijms-19-03888-t001:** Tomato genotypes used in this study.

1. Wild Tomato Species						
Species Name ‘Cultivar’	Accession	Collection Site	Latitude	Longitude	Altitude	Other Comments
*S. corneliomulleri*	LA1274	Lima (Perú)	11°27′36″	76°54′0″	1440 m	Fruits from 6 plants
*S. peruvianum*	LA1336	Arequipa (Perú)				
*S. chilense*	LA1932	Arequipa (Perú)	15°25′0″	74°42′0″	1100 m	Fruits from 15 plants, stress tolerant
*S. huaylasense*	LA1983	Ancash (Perú)	8°4121″	77°58′20″	940 m	Fruits from 1 plant, very dry spot
*S. chmielewskii*	LA2663	Cusco (Perú)	13°41′44″	74°59′39″	2500 m	Fruits form 6–7 plants
*S. arcanum*	LA2157	Cajamarca (Perú)	6°30′21″	78°48′32″	1600 m	Fruits from 2 plants
*S. galapagense*	LA1044	Galapagos Islands (Ecuador)	0°17′4″	90°32′54″	<100 m	
*S. cheesmaniae*	LA1037	Galapagos Islands (Ecuador)	0°25′21″	91°7′0″	800 m	From bottom of volcano crater
*S. pimpinelifollium*	LA1587	La Libertat (Perú)	7°20′0″	79°35′0″	<100 m	Fruits from 20 plants. Grown on river sand
**2. Commercial Tomato Cultivars**		**3. ‘Micro-Tom’ developmental mutants**				
**‘Cultivar name’**	**Accession**	**Mutant**	**Phenotype**	**Gene Product**	**Gene Function**	**References**
‘Ailsa Craig’	LA2838A	*anthocyanin absent* (*aa*)	Anthocyanin deficient	SlGSTAA	Anthocyanin transport	[18,20]
‘Craigella’	LA3247	*anthocyaninless* (*a*)	Low anthocyanin levels	F3’5’H	Anthocyanin biosynthesis	[18,21]
‘Moneymaker’	LA2706	*bushy* (*bu*)	Short internodes	Unknown		[18]
‘Micro-Tom’	LA4480	*lutescent* (*l*)	Premature senescence	Unknown		[18]
		*Never ripe* (*Nr*)	Low ethylene responses	SlETR3	Ethylene receptor	[18,22]
		*sitiens* (*sit*)	ABA deficient	ABA aldehyde oxidase	ABA biosynthesis	[18,23]

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
