# Peer review of "Morphological Characterization of Root System Architecture in Diverse Tomato Genotypes during Early Growth"

_ijms, 2018, doi:10.3390/ijms19123888_

Reviewer 1 Report

This paper describes root architectural features in wild and cultivated tomatos. The data are well presented and of interest . However, the manuscript would benefit from certain restructuring.

The introduction should not end  with a summary of results, but with a clear statement of  the objectives and the working hypothesis.

Results and Discussion section is rather inconnect. For example line 195 onward: First a statement on ABA and than within the same paragraph a switch to flavonoid. There is no clear argumental line.

Paragraph 2.4 provides a nice heat map, but without any further discussion of the results

Usually  conclusions should not contain literature references but refer to the actual results of the presented study. Moreover, outlook to the authors' future studies should be kept to a minimum 

Reviewer 2 Report

In this interesting manuscript Alaguero-Cordovilla and colleagues characterize the root system architecture of different wild and domesticated tomato accessions. In their studies they include also NILs of MICROTOM mutant collection, showing altered hormonal/anthocyanins pathways. They find out that there are differences in the root architectures of the studied accessions such as lateral root distribution and length, adventitious root formation and seed germination. Finally, they quantify and correlate these differences and they include them in a heath map that help to visualize the morphological variability among accessions.

-As the authors report, several molecular mechanisms are involved in the reported morphological differences and these involve most of the times hormones. I think that it would be very interesting (although optional) to know whether auxin or other hormones alter in a different fashion the studied traits in these accessions (e.g. for the LR development part, do all the accessions respond similarly if treated with different auxin concentrations?).

-Authors should clarify from the beginning what they mean for “early traits”

-Authors should explain more in details why they use the NILs of MICROTOM mutant collection.

-I think that it would be extremely useful for the reader if the TABLE S1 would be in the main text, as it would make easier to understand the reported accessions and mutants. 

-Line 74: early germination in comparison to?

-line 106/107 “Significant variations in key RSA parameters, such as LR density, LR length and LR growth

angle, have been reported in different species” this sentence should be rephrased as it is not clear that those are variations among different ecotypes of species that are not tomato.

Reviewer 3 Report

The manuscript submitted by Alaguero-Cordovilla et. al. described a very interesting topic about the morphological characterization of the RSA using different tomato genotypes, which is quite informative and could be used as a good reference for the tomato, even for the other plant species research community. This manuscript is well written, experiments are well designed and nicely performed, data are nicely collected and analyzed. I would like to suggest to consider this piece of work to be published on IJMS. The authors are suggested to uniform the styles in the reference before its final acceptance. 
